# Exercise Training Enhances Platelet Mitochondrial Bioenergetics in Stroke Patients: A Randomized Controlled Trial

**DOI:** 10.3390/jcm8122186

**Published:** 2019-12-11

**Authors:** Chih-Chin Hsu, Hsing-Hua Tsai, Tieh-Cheng Fu, Jong-Shyan Wang

**Affiliations:** 1Department of Physical Medicine and Rehabilitation, Keelung Chang Gung Memorial Hospital, Keelung 204, Taiwan; steele0618@gmail.com (C.-C.H.); fic6481@gmail.com (T.-C.F.); 2School of Medicine, College of Medicine, Chang Gung University, Taoyuan 333, Taiwan; 3Healthy Aging Research Center, Graduate Institute of Rehabilitation Science, College of Medicine, Chang Gung University, Taoyuan 333, Taiwan; serena751004@gmail.com; 4Research Center for Chinese Herbal Medicine, College of Human Ecology, Chang Gung University of Science and Technology, Taoyuan 333, Taiwan

**Keywords:** stroke, exercise training, platelet, mitochondria

## Abstract

Exercise training (ET) may impact physical fitness by affecting mitochondrial functions. This study aimed to elucidate the effect of ET on aerobic capacity and platelet mitochondrial bioenergetics (MTB) in stroke patients. Among the 30 stroke patients who underwent the traditional rehabilitation program (TRP), 15 were randomly assigned to have ET (50–60% VO_2peak_ for 30 min/day, 5 days/week for 4 weeks), and those remaining received only the TRP (control group). The peak exercise capacity (VO_2peak_) and platelet MTB, including oxidative phosphorylation (OXPHOS) and the electron transport chain (ETC), were measured through automatic gas analysis and high-resolution respirometry, respectively. The results demonstrated that ET significantly increased the VO_2peak_ (17.7%) and O_2_ uptake efficiency slope (31.9%) but decreased the ventilation versus CO_2_ production slope (−7.65%). Patients who underwent ET also had significantly enhanced platelet mitochondrial OXPHOS and ETC by activating the FADH2 (Complex II)-dependent pathway, but depressed plasma myeloperoxidase (−28.4%) and interleukin-6 levels (−29.9%). Moreover, changes in VO_2peak_ levels were positively correlated with changes in platelet OXPHOS and ETC capacities. In conclusion, ET increases the platelet MTB by enhancing Complex II activity in stroke patients. The exercise regimen also enhances aerobic fitness and depresses oxidative stress/pro-inflammatory status in stroke patients.

## 1. Introduction

Stroke, caused by cerebral vascular atherosclerosis/thrombosis or intracranial hemorrhage, is the leading cause of long-term disability, which produces enormous global health and economic burdens [1,2]. Stroke-induced functional impairment reduces exercise capacity, which negatively affects the ability of patients to perform their daily activities. This limitation further decreases their independence and quality of life [1,2]. Although a randomized clinical study for 61 chronic stroke patients showed exercise training (ET) improved cardiovascular fitness [3] and a scientific overview encouraged post-stroke survivors to consider the importance of ET in regaining activities of daily living [4], controversy persists regarding how ET influences the hemostasis in stroke patients or stroke prevention [1,5].

Platelets play a pivotal role in thrombogenesis of stroke patients [6,7]. Platelet mitochondria are directly involved in the cellular redox balance, activation, and apoptosis, thereby modulating thrombogenesis [8,9,10]. Regular exercise improves physical performance and aerobic capacity, which is concurrent with reducing the risk of major vascular thrombotic events [11]. According to our previous studies, moderate-intensity ET attenuated platelet reactivity at rest and during strenuous exercise [12,13]. Recent investigations further demonstrated that high-intensity interval training improved the platelet mitochondrial bioenergetics, which might reduce the risk of thrombosis in healthy sedentary individuals [14] and heart failure (HF) patients [15]. However, the effects of ET on platelet mitochondrial bioenergetics in stroke patients have not been extensively reviewed.

A common method to measure mitochondrial bioenergetics is to determine the enzymatic activity of each complex in the electron transport chain (ETC) [16]. However, mitochondria do not work as independent units [17]. Electron transport complexes are interconnected on the mitochondrial inner membrane and turn into respiratory supercomplexes or respirasomes [18]. A polarographic oxygen sensor can measure the mitochondrial respiration of intact cells [19]. Furthermore, individual complexes of the ETC can also be surveyed by addition of exogenous substrates and inhibitor into permeabilized cells [20]. In order to determine the capacities of platelet mitochondria oxidative phosphorylation (OXPHOS) and ETC in stroke patients, the development of novel protocols to evaluate ET effects on mitochondrial OXPHOS and ETC activities are needed.

We hypothesized that ET would improve the hemostasis by enhancing the platelet mitochondria function, which would further result in the increase of exercise capacity in stroke patients. The present study assessed how 4 weeks of moderate-intensity ET (50–60% of VO_2peak_) affected systemic aerobic capacity and platelet mitochondrial oxidative phosphorylation (OXPHOS), as well as ETC activities in stroke patients. With these in-depth analyses, an effective exercise regimen for stroke patients can possibly be established.

## 2. Methods

### 2.1. Participants

A total of 52 stroke patients diagnosed by the neurologist were surveyed from August 2017 to July 2018 in a tertiary care hospital. Brain computed tomography was used to diagnose the hemorrhagic or ischemic stroke. The ischemic stroke subtype was determined according to the TOAST classification system [21]. The inclusion criteria were listed as follows: (I) ≥20 years old; (II) stable stroke events ≥3 months; (III) mini-mental state examination (MMSE) >24; (IV) no acute coronary syndrome; (V) lower extremity function > Brunnstrom stage III with active voluntary movement. Those who had unstable angina, systolic blood pressure >200 mmHg or diastolic blood pressure >110 mm Hg, symptomatic orthostatic hypotension, severe aortic stenosis (peak systolic pressure gradient >50 mmHg, or an aortic valve opening area <0.75 cm^2^), inflammatory disease within the previous 3 months, uncontrolled cardiac dysrhythmias, uncompensated heart failure (HF), third-degree atrioventricular block, pericarditis or myocarditis within the previous 3 months, embolic disease within the previous 3 months, ST segment displacement ≥2 mm at rest, or uncontrolled diabetes (blood glucose ≥300 mg/dL or ≥250 mg/dL with ketone bodies) were excluded from the study.

All our eligible candidates did not engage in regular physical exercise except 30–45 min of supervised in-hospital traditional rehabilitation programs (TRP), which included balance, neurofacilitation, and therapeutic exercise, as instructed by their rehabilitation physicians, before starting intervention. They were randomly assigned to the intervention (under ET and TRP) and control (under TRP only) groups (Appendix A) based on the computer-generated random allocation sequence. An independent physician performed the randomization procedure, which was concealed to the members performing the intervention and the enrolled participants before starting the experiment. All subjects provided informed consent after the experimental procedures were explained. The Institutional Review Board of a tertiary care hospital approved this study (IRB No. 201602058A3) and the clinical trial registration number was NCT03960918.

### 2.2. Graded Exercise Test (GET)

The physician performing the GET was blinded to the subject grouping. He assessed their cardiopulmonary fitness 2 days before and 2 days after 4-week intervention. Moreover, the data collector was isolated from the analytic specialist. Each participant underwent the GET at an incremental work-rate of 10 W/min on a bicycle ergometer (Ergoselect 150P, ergoline GmbH, Bitz, Germany) with a continuous monitoring heart rate (HR), brachial blood pressure, and arterial O_2_ saturation, until the stopping conditions described previously [13,14]. O_2_ consumption (VO_2_), minute ventilation (V_E_), and carbon dioxide production (VCO_2_) were measured breath by breath by a cardiopulmonary measurement device (MasterScreen CPX, CareFusion Corp., Hoechberg, Germany), and the VO_2peak_ was defined as the guideline for exercise testing suggested by the American College of Sports Medicine (ACSM) [22].

V_E_ and VCO_2_ responses acquired from the initiation of exercise to the peak values were used to calculate the V_E_-VCO_2_ slope using least-square linear regression [23]. The O_2_ uptake efficiency slope (OUES), an estimation of the efficiency of O_2_ consumption during exercise, was derived from the slope of a natural logarithm plot of V_E_ vs. VO_2_. A greater slope indicated a higher ventilation efficiency [23,24]. Additionally, a 6-min walk test (6 MWT) was used to screen functional capacity and exercise endurance in stroke patients [25].

### 2.3. Cardiac Hemodynamic Measurements

A noninvasive continuous cardiac output monitoring system (NICOM, Cheetah Medical, Wilmington, DE, USA) was used to evaluate cardiac hemodynamic response to exercise, which analyzes the phase shift (ΔΦ) created by alternating electrical current across the chest of the subject, as described in our previous study [26].

### 2.4. Health-Related Quality of Life

Quality of life (QoL) was measured using a Short Form-36 Health Survey questionnaire (SF-36), the findings of which were related to the co-morbidities of stroke patients [27].

### 2.5. Exercise Training (ET) Protocol

A 30–45 min of TRP was implemented for all included stroke survivors, which was judged by physician discretion. The control subjects only engaged in the TRP. In addition to the TRP, the subjects in the intervention group underwent supervised hospital-based training 20 times (5 session/week for 4 weeks) on a bicycle ergometer (Ergoselect 150P, Bitz, Germany) as our previous protocol describes [26]. The ET comprised a warm-up at 30% of VO_2peak_ for 3 min, followed by continuous 60% of VO_2peak_ for 30 min, then a cool-down at 30% of VO_2peak_ for 3 min. The training was terminated when the subject had symptoms/signs suggested by the ACSM guideline [22]. The compliance rate of all our subjects was 100% (Appendix A).

### 2.6. Blood Cell Count

Based on our previous work [28], an amount of 25 mL of blood was sampled just prior to each exercise test performed. The first 2 mL was discarded, then the remaining blood sample was used for the measurements of hematological parameters and platelet function. Each blood sampling was performed after holding medication (anti-platelet or anticoagulant) for 48 h, stroke prevention medication was restarted after the procedure. The blood analysis was performed initially and after 4 weeks of ET. Blood cells were enumerated using a Sysmax SF-3000 cell counter (GMI Inc., Ramsey, MN, USA) and were repetitively analyzed twice to ensure the reproducibility of the results.

### 2.7. Platelet Isolation

Samples, processed as the above description to avoid platelet activation [28], were collected in polypropylene tubes containing a sodium citrate concentration of 3.8 g/dL. Platelet rich plasma (PRP) was prepared through centrifugation at 300× *g* for 10 min at approximately 20 °C. Platelets were sedimented through centrifugation of the PRP at 1500× *g* for 10 min at approximately 20 °C and then were washed once with PBS containing 4 mM ethylenediaminetetraacetic acid (EDTA) (Sigma-Aldrich, St. Louis, MO, USA) to inhibit platelet activation [14,15]. They were mixed with mitochondrial respiration medium (MiR05, containing sucrose 110 mM, HEPES 20 mM, taurine 20 mM, K-lactobionate 60 mM, MgCl_2_ 3 mM, KH_2_PO_4_ 10 mM, EGTA 0.5 mM, BSA 1 g/L, pH 7.1) to a final concentration of 2 × 10^8^ cells/mL. All platelet fractions were analyzed within 2 h after cell purification.

### 2.8. High-Resolution Respirometry

Platelet mitochondrial respiration was measured by a high-resolution respirometry (Oxygraph-2K, Oroboros Instrument, Innsbruck, Austria) with a stirrer speed of 750 rpm at a constant temperature of 37 °C. Data were acquired and recorded every 2 s by DatLab software version 6 (Oroboros Instrument, Innsbruck, Austria). Platelets of 2 × 10^8^ cells/mL were added to the glass chamber filled with 2 mL mitochondrial respiration medium (MiR05) for measurement [14,15]. The O_2_ concentration was automatically calculated from barometric pressure and the solubility factor was 0.92 for MiR05.

### 2.9. Mitochondrial ETC and OXPHOS in Platelets

The substrate-uncoupler-inhibitor titration reference protocol (SUIT-RP), consisting of two mitochondrial substrate-controlled experiments (RP1 and RP2), was applied to acquire the platelet mitochondrial bioenergetics (Appendix A). All the chemicals were purchased from the Sigma-Aldrich Corporation.

The SUIT-RP1 was used to measure the mitochondrial ETC capacity (Appendix A). The cell membrane was permeabilized with digitonin. Malate (2 mM) and pyruvate (5 mM), NADH-linked (N-linked) substrates were subsequently added [14,15]. O_2_ consumption was only driven by uncoupling proton leakage (LEAK state, PM_L_) because the ADP was absent. The OXPHOS capacity (PM_P_) was then evaluated by 1 mM ADP (Calbiochem, St. Louis, MO, USA) titration. The ETC capacity driven by malate and pyruvate (PM_E_) was obtained by mitochondrial protonophore carbonyl cyanide-p-trifluoromethoxyphenylhydrazone (FCCP) titration (0.5 μM/step) until no further respiration increased. Glutamate (10 mM) was added to access the maximal ETC capacity driven by NADH or mitochondrial complex I (CI)-related resources (MPG_E_). Thereafter, 10 mM succinate was used to complete the convergent input of CI and complex II (CII) (MPGS_E_). Octanoyl-carnitine (Oct) titration (0.5 mM) was performed to evaluate the additional effect of fatty acid oxidation (FAO) (MPGSOct_E_). The N-linked substrate-dependent respiration and FAO pathways (S_E_) were blocked by 0.5 μM rotenone (CI inhibitor). The additional contribution of the mitochondrial complex of glycerophosphate dehydrogenase (CGpDH) was measured with 10 mM glycerophosphate (SGp_E_). At last, the mitochondrial respiration was inhibited by 2.5 μM antimycin A, the mitochondrial complex III inhibitor (Appendix A).

The mitochondrial ETC capacities in the permeabilized platelets were obtained from the following equations:
ETC_CI_ (PMG_E_) = Pyruvat + Malate + ADP + FCCP + Glutamate(1)
ETC_CI+CII_ (PMGS_E_) = Pyruvate + Malate + ADP + FCCP + Glutamate + Succinate(2)
ETC_CI+CII+FAO_ (PMGSOct_E_) = Pyruvat + Malate + ADP + FCCP + Glutamate + Succinate + Oct(3)
ETC_CII_ (S_E_) = Pyruvate + Malate + ADP + FCCP + Glutamate + Succinate + Oct + Rotenone(4)
ETC_CII+Gp_ (SGp_E_) = Pyruvate + Malate + ADP + FCCP + Glutamate + Succinate + Oct + Rotenone + Glycerophosphate(5)


The SUIT-RP2 was to measure the ATP synthase-dependent OXPHOS state (Appendix A). Following routine respiration and digitonin permeabilization, 1 mM ADP was titrated to accelerate the depletion of residual endogenous substrates. Then, 0.5 mM OC and 0.1 mM malate were added to evaluate the FAO. An amount of 2 mM Malate was used to reduce the flux from succinate dehydrogenase. This procedure could also activate the mitochondrial malic enzyme and led to higher O_2_ consumption (OctM_P_). An amount of 5 mM pyruvate (OctMP_P_) and 10 mM glutamate (OctMPG_P_) were sequentially titrated to acquire the N-linked respiration. A consecutive titration of 10 mM succinate and 10 mM glycerophosphate stimulated the mitochondria respiration through CII (OctMPGS_P_) and CGpDH (OctMPGSGp_P_) input. To evaluate the limitation of the OXPHOS system, FCCP (0.5 μM/step) was titrated to access the maximal ETC capacity (OctMPGSGp_E_). An amount of 0.5 μM rotenone was added to reach the identical SGp_E_ state in the RP1 and RP2 protocols. Antimycin A (2.5 μM) was added in the final step to block the whole mitochondria respiration.

The capacities of mitochondrial OXPHOS in permeabilized platelets were calculated from the following equations (Appendix A):
OXPHOS_FAO_ (Oct_P_) = ADP + Oct(6)
OXPHOS_FAO+CI_ (OctMPG_P_) = ADP + Oct + Malate + Pyruvate + Glutamate(7)
OXPHOS_FAO+CI+CII_ (OctMPGS_P_) = ADP + Oct + Malate + Pyruvate + Glutamate + Succinate(8)


This study defined the maximal capacities of OXPHOS and ETC in platelets as the OctMPGSGp_P_ and OctMPGSGp_E_ states in the RP2 protocol, respectively (Appendix A).
OXPHOS_FAO+CI+CII+Gp_ (OctMPGSGp_P_) = ADP + Oct + Malate + Pyruvate + Glutamate + Succinate + Glycerophosphate(9)
ETC_FAO+CI+CII+Gp_ (OctMPGSGp_E_) = ADP + Oct + Malate + Pyruvate + Glutamate + Succinate + Glycerophosphate + FCCP(10)


### 2.10. Biomarkers of Oxidative Stress and Pro-Inflammation in Plasma

A 5-mL blood sample was placed in a cold centrifuge tube containing 4 mM EDTA, and was immediately centrifuged at 3000× *g* for 10 min at 4 °C. Plasma myeloperoxidase (MPO) (Immunology Consultants Lab., Newberg, OR, USA) and interleukin-6 (IL-6) (eBioscience, SanDiego, CA, USA) levels were quantified by commercially available ELISA kits [26]. Prominent increases of TNF_α_ and IL-1 occur within few hours in the inflammatory cascade and may be inadequate for the study for chronic stroke patients [29]. Therefore, IL-6 and MPO were reviewed in the study because they have been reported to be associated with chronic low-grade inflammation and good responses to exercise related anti-inflammatory effects [29,30].

### 2.11. Statistical Analysis

Data were expressed as mean ± SEM and were analyzed using the statistical software package StatView. The experimental results were analyzed by 2 (groups) × 2 (time sample points; i.e., pre- and post-interventions) repeated measures ANOVA. Bonferroni’s post hoc was used to compare cardiopulmonary fitness, biomarkers of plasma oxidative stress and inflammation, and platelet mitochondria functions at the beginning of this study and after 4 weeks of observation in both groups. Pearson correlation analysis was used to determine the correlation between the VO_2peak_ and platelet mitochondrial OXPHOS and ETC in stroke patients. *p* values lower than 0.05 were considered statistically significant.

## 3. Results

### 3.1. Cardiopulmonary Fitness and Health-Related QoL

A total of 30 subjects completed the study in the intervention (*n* = 15) and control (*n* = 15) groups (Figure 1). No adverse vascular or thrombotic event occurred in the two groups during the investigation. Differences of baseline anthropometric and clinical parameters between the two groups were not significant (Table 1). Among the included subjects, baseline clinical information and comorbidities of the hemorrhagic stroke patients in the two groups is shown in the Appendix A. All our hemorrhagic stroke patients suffered from hypertension. The initial cardiopulmonary fitness of the two groups was also similar (Table 2).

Following 4 weeks of interventions, the stroke patients underwent ET displayed significantly increased work-rate, heart rate, stroke volume (SV), cardiac output (CO) as well as, V_E_, VCO_2_, OUES, VO_2peak_, and 6 MWT distance (*p* < 0.05). In addition, a significant decrease of V_E_-VCO_2_ slope (*p* < 0.05) was detected in stroke patients with ET. However, significant changes in cardiopulmonary fitness were not identified in those with 4 weeks of traditional rehabilitation training. In our observation, the TRP did not influence indicators of ventilatory efficiency and functional capacity in stroke patients. Detailed information is shown in Table 2.

In health-related QoL, 4 weeks of ET significantly increased the subclass scores of the physical (scores from 40.1 to 44.2, *p* < 0.05) and mental (scores from 36.1 to 44.9, *p* < 0.05) dimensions (Table 2). However, QoL in the stroke patients who underwent 4 weeks of TRP remained unchanged (Table 2).

### 3.2. Hematologic Parameters and Oxidative Stress/Inflammatory Biomarkers

There were no significant changes in hemograms (i.e., erythrocyte, hemoglobin, hematocrit, leukocyte, and platelet) following 4 weeks of TRP with or without ET (Table 1). However, ET associated with TRP considerably reduced the plasma levels of MPO (*p* < 0.05) and IL-6 (*p* < 0.05). The oxidative stress and pro-inflammation in plasma were unchanged following the TRP alone (Table 1).

### 3.3. Mitochondrial ETC and OXPHOS in Platelets

Neither TRP + ET nor TRP alone changed the routine respiration and uncoupling proton leakage of platelets (Figure 1 and Figure 2). Moreover, the two therapeutic regimens were also insufficient to influence platelet ETC driven by NADH (CI)-dependent pathway (PM_E_ and PMG_E_, Figure 1A,B). The OXPHOS driven by NADH (CI)- and FAO-dependent pathways (Oct_p_, OctM_p_, OctMP_p_, and OctMPG_p_) were not affected (Figure 2A,B).

Notably, TRP + ET for 4 weeks significantly increased platelet flavin adenine dinucleotide (FADH2) in CII-related ETC levels (PMGS_E_, PMGSOct_E_, S_E_, and SGp_E_) (Figure 1A, *p* < 0.05). Moreover, changes in VO_2peak_ levels were directly related to changes of PMGS_E_ (Figure 3A, r = 0.587, *p* < 0.001), PMGSOct_E_ (Figure 3B, r = 0.581, *p* < 0.001), S_E_ (Figure 3C, r = 0.650, *p* < 0.001), and SGp_E_ (Figure 3D, r = 0.846, *p* < 0.001).

On the other hand, TRP + ET also enhanced succinate-involved platelet OXPHOS levels (Figure 2A, *p* < 0.05), such as OctMPGS_P_, rather than OctMPG_P_. Furthermore, the exercise regimen also augmented the capacities of the maximal OXPHOS (OctMPGSGp_P_, *p* < 0.05) and ETC (OctMPGSGp_E_, *p* < 0.05) in platelet (Figure 2A). Additionally, changes in VO_2peak_ levels were positively associated with changes in OctMPGSGp_P_ (Figure 4A, r = 0.587, *p* < 0.001) and OctMPGSGp_E_ in platelet (Figure 4B, r = 0.581, *p* < 0.001). However, no significant changes in platelet mitochondrial bioenergetics, including ETC and OXPHOS capacities, were observed in the control subjects (Figure 1 and Figure 2).

## 4. Discussion

The present work exhibited that 4 weeks of moderate-intensity ET improved exercise capacity and ventilatory/hemodynamic efficiency, and physiological adaptation was accompanied by reduced oxidative stress and pro-inflammatory status in stroke patients. The ET-induced relief of oxidative stress in both ischemic and hemorrhagic stroke patients in the study might reverse the systemic coagulopathy and may be helpful in ischemic and hemorrhagic stroke prevention. In addition to the above interesting clinical findings, two novel reference protocols (SUIT-RP1 and SUIT-RP2) were developed in this study to determine the platelet mitochondrial OXPHOS and ETC capacities. To our knowledge, this human study is the first to demonstrate that moderate-intensity ET effectively enhanced platelet mitochondrial OXPHOS and ETC capacities through increasing Complex II activity in stroke patients.

### 4.1. Exercise Capacity

In comparison with healthy individuals of the same age, reduced exercise capacity in stroke patients, especially about one month after onset of stroke, decreased their ability to perform daily activity independently, thereby further worsening their QoL [4,5]. The ventilatory parameters obtained from the GET may convey information regarding prognosis of circulatory disorders [23]. The VO_2peak_ and OUES are indicators of cardiorespiratory fitness and O_2_ metabolic efficiency, respectively [22,23]. The V_E_-VCO_2_ slope has long been known to be a useful survival predictor in patients with circulatory disorders [23]. Our previous study reported that these indices of ventilatory efficiency were correlated with exercise-induced central and peripheral hemodynamic changes in HF patients. We also found that ET effectively improved ventilation–perfusion matching during exercise, and this improvement was accompanied by better QoL in HF patients [25]. In this study, VO_2peak_ and OUES, as well as the cardiac pumping capability (such as CO and SV) of stroke patients, improved significantly after 4 weeks of moderate-intensity ET. These post-exercise benefits associated with a decreased trend in the V_E_-VCO_2_ slope suggest that ET effectively improves ventilatory and hemodynamic efficiencies, leading to an increase of exercise capacity in stroke patients. In addition, the exercise regimen also increased the capacities of the maximal platelet OXPHOS and ETC in stroke patients. Moreover, improved VO_2peak_ was positively correlated with platelet mitochondrial bioenergetics in the stroke patients. Hence, platelet mitochondrial bioenergetics may be an ideal surrogate for determination of systemic aerobic capacity in stroke patients.

### 4.2. Mitochondrial Functionalities in Platelets

Emerging findings indicate that neuron death in stroke patients is associated with an impaired mitochondria respiratory capacity, which may be caused by the inability to maintain energy homeostatic to compensate for the acute cerebrovascular event. Oxygen deprivation after stroke onset contribute to mitochondria dysfunction, resulting in overproduction of reactive oxidative species (ROS) [31]. Systemic inflammatory response occurs subsequently and further changes of pro- or anti- inflammatory cytokines are associated with the severity of neurological deficit [32]. Additionally, mitochondria play a pivotal role in platelet-mediated aggregation and are essential in non-ATP- mediated thrombosis [8,9]. Hence, platelet mitochondrial bioenergetics can be a marker reflecting the oxidative stress of stroke.

Since mitochondria are highly sensitive and respond dynamically to oxidative stress in their microenvironment [33], mitochondria dysfunctions are frequently observed in the early stage of a typical ischemia/reperfusion tissue injury, such as acute myocardial infarction or stroke. Overproduction of mitochondria-derived ROS has been considered to be the consequence of the interaction of the malfunctional respiratory chain with O_2_ during reperfusion [34]. The present investigation demonstrated that moderate-intensity ET significantly decreased plasma MPO and IL-6 levels, inflammatory cytokines, in stroke patients. Thus, it is considered that the ET may blunt stroke-induced oxidative stress possibly incited by mitochondrial dysfunction.

The accumulation of intracellular succinate after an ischemic insult, resulting in elevated mitochondrial ROS production, has been reported [35]. Oxidative stress may further induce succinate overproduction by decreasing succinate dehydrogenase (SDH) activity [36]. Our previous study on sedentary healthy young adults found that ET under O_2_ tension of 12% for 30 min significantly enhanced platelet mitochondria SDH activity and Complex II-related respiration [14]. In another clinical investigation, we further identified that ET elevated the platelet mitochondrial O_2_ consumption rate against oxidative stress by increasing Complex II activity in HF patients [14]. In the present study, 4 weeks of ET enhanced platelet mitochondrial OXPHOS and ETC capacities by increasing Complex II activity, but lowered plasma MPO and IL-6 concentrations in the stroke patients. In this investigation, we further detected that an improved aerobic capacity was positively correlated with changes in FADH2 (CII)-related OXPHOS and ETC activities. Therefore, ET-induced elevation in platelet Complex II activity may rapidly eliminate succinate and further reduce ROS production from platelet mitochondria. This ET-induced physiologic adaptation may eventually depress circulatory oxidative stress and pro-inflammatory status in stroke patients.

### 4.3. Health-Related Quality of Life

In addition to an increase in aerobic capacity, this study demonstrated that ET significantly improved mental health in stroke patients. These findings imply that the moderate-intensity exercise regimen effectively facilitated the ability of patients to cope with the physical demands of daily activity, subsequently improving psychosocial status in stroke patients. Furthermore, an improved health-related QoL might improve survival probability in stroke patients and simultaneously reduce the financial burden in their health care system [1,2].

### 4.4. Limitations of This Study

Although limited stroke patients were included in the study, the results regarding platelet mitochondrial bioenergetics have high values of statistical power ranging from 0.902 to 1.000 (Appendix A). Additionally, this study mainly focused on the effects of ET on platelet mitochondrial bioenergetics rather than platelet reactivity. Our previous studies investigated the effect of different aerobic exercise regimens on underlying mechanisms of platelet reactivity in healthy people and patients with cardiovascular disorders [11,12,13]. A recent study further reported that aerobic exercise training markedly suppressed the hypoxia-induced oxidative damage of platelet mitochondria and consequently, reduced platelet-mediated thrombin generation in healthy sedentary individuals [14]. However, the role of ET-mediated improvement of platelet mitochondrial function on platelet reactivity and coagulation in stroke patients requires further investigation.

Six hemorrhagic strokes, 3 in each group, suffered from hypertension in this study. It is well-known that hypertension is highly prevalent and an important risk factor for hemorrhagic stroke [37]. Increased platelet aggregation has been observed in aneurysmal subarachnoid hemorrhagic patients and was possibly related with abnormal platelet ADP secretion [38]. Moreover, elevated oxidative stress may impair mitochondria function and could be a potent trigger of consumptive coagulopathy to result in abnormal cerebral hemostasis [39]. Therefore, we recruited hemorrhagic stroke patients because both hypertension and abnormal platelet function can be expected in them.

## 5. Conclusions

In the present study, 4 weeks of ET improved systemic aerobic capacity through increased ventilatory and hemodynamic efficiencies in stroke patients. The exercise regimen also increased the capacities of OXPHOS and ETC by enhancing mitochondrial Complex II activity in platelets. These experimental findings may provide insight into the identification of adequate exercise regimen to increase physical performance and improve platelet mitochondrial bioenergetics in stroke patients.

## Figures and Tables

**Figure 1 jcm-08-02186-f001:**
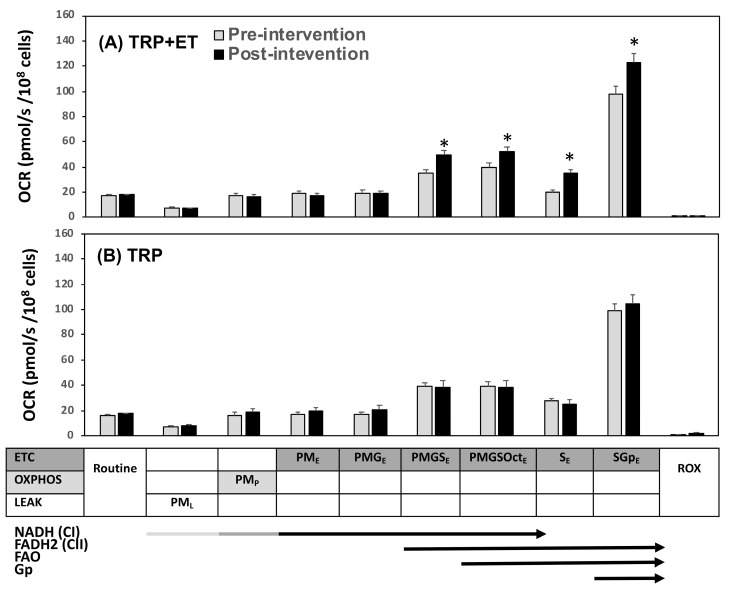
Effects of traditional rehabilitation program (TRP) with exercise training (ET) (**A**, TRP + ET) and without ET (**B**, TRP) in platelet mitochondrial O_2_ consumption rate (OCR) using the reference protocol 1 (RP1) protocol in patients with cerebral vascular accident. P, pyruvate; M, malate; G, glutamate; S, succinate; Oct, octanoyl-carnitine; Gp, glycerophosphate; Rot, rotenone; Ama, antimycin; ROX, residual O_2_ consumption; OXPHOS or _P_, oxidative phosphorylation; ETC or _E_, electron transport system; LEAK or _L_, uncoupling proton leakage. Pre-intervention vs. post-intervention, * *p* < 0.05. Values were mean ± SEM.

**Figure 2 jcm-08-02186-f002:**
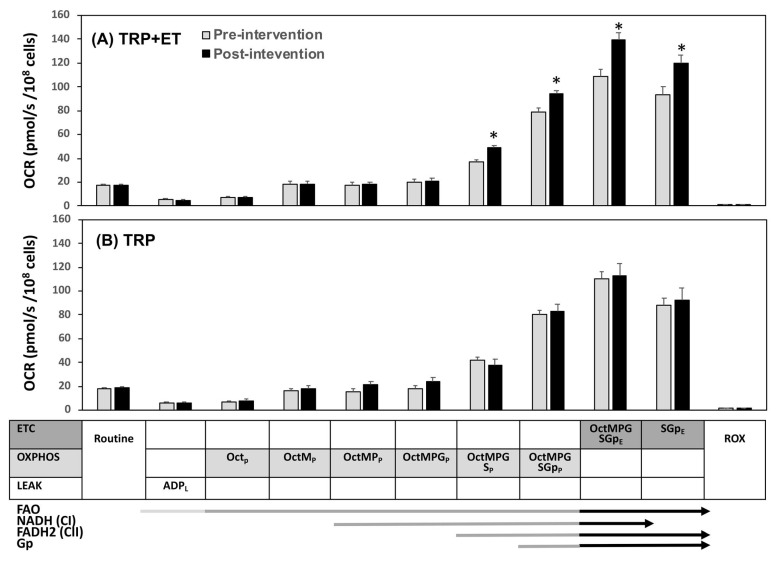
Effects of TRP with (**A**, TRP + ET) and without (**B**, TRP) exercise training in platelet mitochondrial O_2_ consumption rate (OCR) using the reference protocol 2 (RP2) protocol in patients with cerebral vascular accident. P, pyruvate; M, malate; G, glutamate; S, succinate; Oct, octanoyl-carnitine; Gp, glycerophosphate; Rot, rotenone; Ama, antimycin; ROX, residual O_2_ consumption; OXPHOS or _P_, oxidative phosphorylation; ETC or _E_, electron transport system; LEAK or _L_, uncoupling proton leakage. Pre-intervention vs. post-interve ntion, * *p* < 0.05. Values were mean ± SEM.

**Figure 3 jcm-08-02186-f003:**
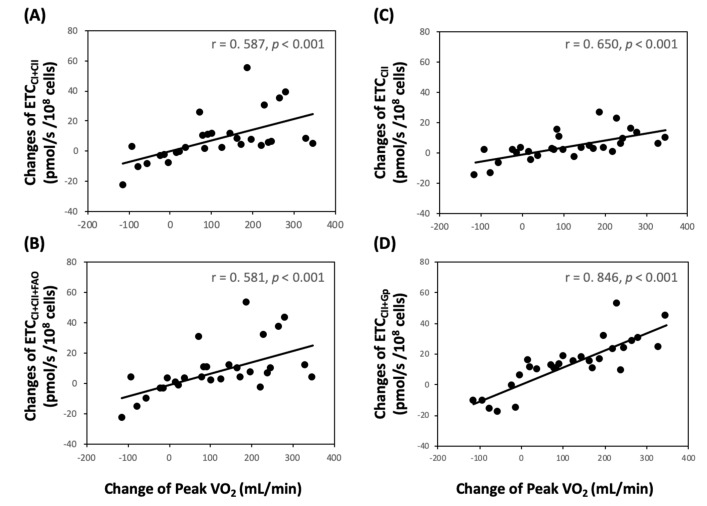
Correlations between changes of aerobic capacity (VO_2peak_) and platelet mitochondrial electron transport chain (ETC) in patients with cerebral vascular accident. (**A**) ETC_CI+CII_ (PMGS_E_) = Pyruvate + Malate + ADP + FCCP + Glutamate + Succinate; (**B**) ETC_CI+CII+FAO_ (PMGSOct_E_) = Pyruvate + Malate + ADP + FCCP + Glutamate + Succinate + Octanoyl-Carnitine; (**C**) ETC_CII_ (S_E_) = Pyruvate + Malate + ADP + FCCP + Glutamate + Succinate + Octanoyl-Carnitine + Rotenone; (**D**) ETC_CII+Gp_ (SGp_E_) = Pyruvate + Malate + ADP + FCCP + Glutamate + Succinate + Octanoyl-Carnitine + Rotenone + Glycerophosphate.

**Figure 4 jcm-08-02186-f004:**
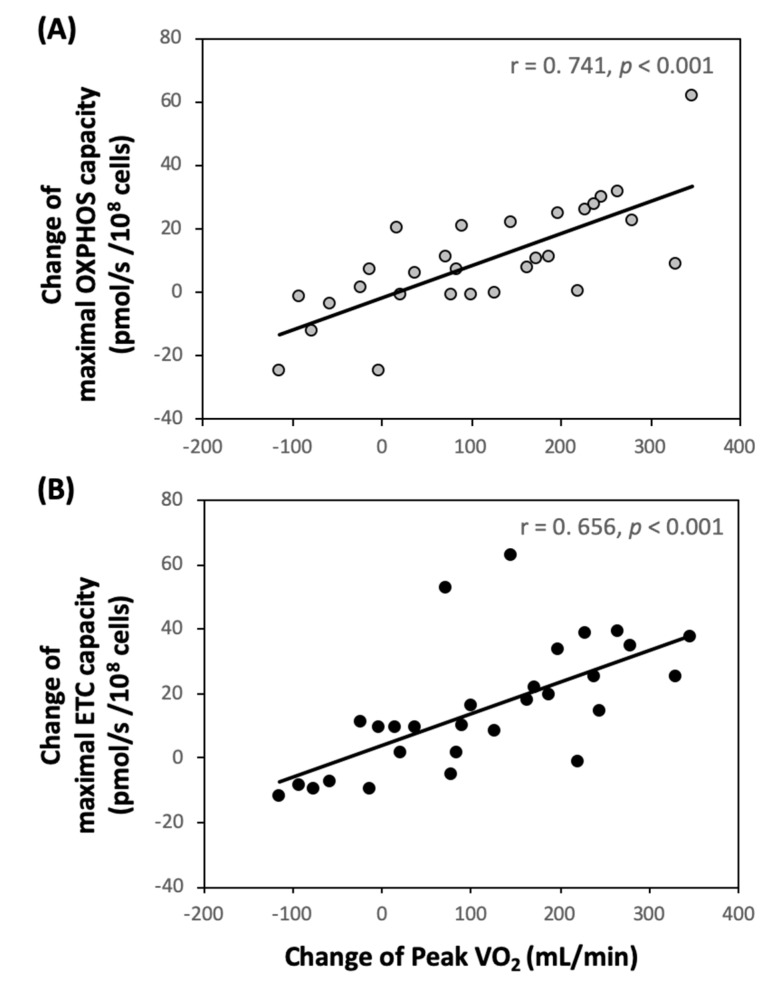
Correlations between changes of aerobic capacity (VO_2peak_) and the capacities of maximal OXPHOS and ETC in platelets (as the OctMPGSGp_P_ and OctMPGSGp_E_ states using RP2 protocol, respectively) in patients with cerebral vascular accident. (**A**) OXPHOS_FAO+CI+CII+Gp_ (OctMPGSGp_P_) = ADP + Octanoyl-Carnitine + Malate + Pyruvate + Glutamate + Succinate + Glycerophosphate; (**B**) ETC_FAO+CI+CII+Gp_ (OctMPGSGp_E_) = ADP + Octanoyl-carnitine + Malate + Pyruvate + Glutamate + Succinate + Glycerophosphate + FCCP.

**Table 1 jcm-08-02186-t001:** Demographic and clinical characteristics in stroke patients.

		TRP + ET		TRP	
		Pre	Post	Pre	Post
Anthropometrics/Clinical Characteristics					
Gender	*n* (M/F)	15 (12/3)	15 (12/3)	15 (13/2)	15 (13/2)
Age	year	55.7 ± 3.0	-	57.8 ± 3.9	-
Height	cm	165.7 ± 1.4	165.7 ± 1.4	166.8 ± 1.5	166.8 ± 1.5
Weight	kg	70.7 ± 2.7	70.0 ± 2.6	71.6 ± 2.8	72.2 ± 2.9
BMI	kg/m^2^	25.8 ± 1.0	25.6 ± 0.9	25.7 ± 1.0	25.9 ± 1.2
Heart rate	bpm	76 ± 5	73 ± 6	77 ± 6	76 ± 7
Systolic blood pressure	mmHg	135 ± 10	132 ± 9	137 ± 9	136 ± 8
Diastolic blood pressure	mmHg	78 ± 4	75 ± 6	81 ± 5	80 ± 6
Etiology					
Ischemia	*n* (%)	12 (80)	-	12 (80)	-
Small vessel thrombosis		7 (47)		8 (53)	
Embolic		5 (33)		4 (27)	
Hemorrhage	*n* (%)	3 (20)	-	3 (20)	-
Stroke duration	month	21 ± 3	-	23 ± 4	-
Brunnstrom Stage					
Stage III	*n* (%)	3 (25)	-	4 (27)	-
>Stage III	*n* (%)	12 (75)	-	11 (73)	-
Mini-Mental State Examination					
	score	28.7 ± 2.9	-	29.2 ± 4.5	-
Risk Factors					
Smoking	*n* (%)	6 (40)		6 (40)	
Hyperlipidemia	*n* (%)	12 (80)	-	11 (73)	-
Hypertension	*n* (%)	3 (20)	-	4 (27)	-
CVD	*n* (%)	7 (47)	-	8 (53)	-
Diabetes mellitus	*n* (%)	10 (67)	-	9 (60)	-
Hematological Parameters					
Erythrocyte	10^6^/μL	5.00 ± 0.12	5.17 ± 0.14	4.91 ± 0.11	4.90 ± 0.16
Hemoglobin	g/dL	14.3 ± 0. 4	14.8 ± 0.5	14.2 ± 0. 4	14.2 ± 0.7
Hematocrit	%	44.6 ± 1.3	45.7 ± 1.4	44.3 ± 1.3	43.1 ± 1.9
Leukocyte	10^3^/μL	7.01 ± 0.21	6.72 ± 0.40	7.17 ± 0.26	6.37 ± 0.53
Platelet	10^3^/μL	264 ± 13	239 ± 16	274 ± 15	255 ± 12
Plasma Oxidative Stress and Pro-Inflammatory Status					
MPO	ng/mL	11.70 ± 0.64	8.38 ± 1.05 *	11.10 ± 0.98	12.15 ± 0.82
IL-6	pg/mL	13.69 ± 1.75	9.61 ± 1.75 *	13.09 ± 1.84	13.03 ± 1.73
Medicines					
ASA	*n* (%)	7 (47)	7 (47)	8 (53)	8 (53)
NOAC	*n* (%)	5 (33)	5 (33)	4 (27)	4 (27)
HMG CoA reductase inhibitor	*n* (%)	12 (80)	12 (80)	11 (73)	11 (73)
β-blockers	*n* (%)	3 (20)	3 (20)	4 (27)	4 (27)
ACEI/ARB	*n* (%)	3 (20)	3 (20)	4 (27)	4 (27)

Values are mean ± SEM. ACEI/ARB, angiotensin converting enzyme inhibitor/angiotensin II receptor blocker; ASA, acetylsalicylic acid; BMI, body mass index; CVD, cardiovascular diseases; ET, exercise training; HMG CoA, 3-hydroxy-3-methyl- glutaryl coenzyme A; IL-6, interleukin-6; MPO, myeloperoxidase; NOAC, non-vitamin K antagonist oral anticoagulant; Pre, pre-intervention; Post, post-intervention; TRP, traditional rehabilitation program. * estimated by repeated measured ANOVA.

**Table 2 jcm-08-02186-t002:** The effect of exercise training on cardiopulmonary responses to exercise and health-related quality of life in stroke patients.

		TRP + ET		TRP	
		Pre	Post	Pre	Post
Peak Exercise Performance					
Work-rate	watt	81.9 ± 4.8	100.0 ± 6.1 *	78.8 ± 7.8	80.5 ± 6.9
Hear rate	bpm	132 ± 4	139 ± 3 *	133 ± 4	134 ± 5
Stroke volume	mL	68.1 ± 3.2	80.5 ± 4.1 *	70.8 ± 3.5	75.7 ± 3.7
Cardiac output	L/min	8.97 ± 0.41	11.16 ± 0.79 *	9.42 ± 0.52	10.15 ± 0.91
V_E_	L/min	50.3 ± 3.1	62.5 ± 3.7 *	51.2 ± 4.1	53.4 ± 5.2
VO_2_	mL/min	1155 ± 80	1359 ± 85 *	1165 ± 76	1194 ± 65
VCO_2_	mL/min	1291 ± 87	1590 ± 97 *	1348 ± 92	1395 ± 95
MET	score	4.7 ± 0.2	5.5 ± 0.2 *	4.6 ± 0.3	4.8 ± 0.2
OUES	unit	630 ± 38	831 ± 58 *	634 ± 61	575 ± 36
V_E_-VCO_2_ slope	unit	35.3 ± 1.9	32.6 ± 2.2	34.6 ± 1.9	34.3 ± 1.5
6-min walk test	meter	325.4 ± 76.2	380.4 ± 47.4 *	320.5 ± 62.1	315.6 ± 56.3
Short Form-36 Health Survey Questionnaire					
Physical	score	40.1 ± 1.7	44.2 ± 2.0 *	41.5 ± 2.1	40.6 ± 2.2
Mental	score	36.1 ± 2.1	44.9 ± 2.2 *	36.8 ± 2.2	34.8 ± 2.7

Values are mean ± SEM. ET, exercise training; TRP, traditional rehabilitation program; Pre, pre-intervention; Post, post-intervention; MET, metabolic equivalent of task; VO_2_, O_2_ consumption; OUES, O_2_ uptake efficiency slope; V_E_, minute ventilation; VCO_2_, CO_2_ production; * estimated by repeated measured ANOVA.

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
