# Peer review of "Exercise Training Enhances Platelet Mitochondrial Bioenergetics in Stroke Patients: A Randomized Controlled Trial"

_jcm, 2019, doi:10.3390/jcm8122186_

Round 1

Reviewer 1 Report

The authors have taken into account all the comments/suggestions and as a result, the quality of the manuscript has been improved.

Author Response

Pleas see the attached file.

Reviewer 2 Report

Thank you for the response to reviewer comments which the authors have respondend well to.

Two minor issue still remains:

Which stroke classification was applied for assessing stroke subtype? Does embolism cover only cardioembolic stroke? How about large artery, did any of the patients suffer from large artery stroke? Please provide details. The paragraph on inclusion of intracerebral hemorrhagic patients is a bit too speculative to justify iniclusion. The most common cause of hemorrhagic stroke is hypertension and atherosclerosis, and only to a lesser degree abnormal platelet function unless NOAC og antiplatelets was used. Please modify the discussion , and I would suggest to include cause of hemorrhage in the three included patients in the results.

Author Response

This manuscript is a resubmission of an earlier submission. The following is a list of the peer review reports and author responses from that submission.

Round 1

Reviewer 1 Report

The study was aimed to study whether Exercise training (ET) could improve hemostasis through inducing an improvement in platelet mitochondrial function, which could consequently increase the capacity of ET in patients suffering from stroke.

Reviewer comments

Introduction

Overall, the introduction section is well written and easy to understand. However, there are certain issues that the authors should address.

Lines 42-45: Even though Exercise training is considered by the authors as a “valuable non-pharmacological intervention”, no further information is provided. I feel that a little explanation regarding what ET is should be included.

Line 56: Electron transport system sounds inaccurate. The authors should replace it for electron transport chain ETC (in this line and throughout the rest of the text).

Line 63: Please change “indicated” for “needed”.

Material and methods

This part of the manuscript is well written and enough information is provided regarding the used methods to analyze the parameters of interest.

Results

Table2: The differences found in the TRP+ET group regarding VE-VCO2 slope between pre and post intervention seems to small to find significant changes (specially taking into account the SEM in the post TRP+ET, as well as the n per group).  

Discussion

The authors studied the levels of MPO and IL-6. However, along with IL-6, IL-1 and TNF-alpha are usually studied in order to assess the oxidative stress status. The authors should include this information or justify why these parameters were not determined.

Lines 353-355: The authors state that ET promoted OXPHOS efficiency in platelet mitochondria without affecting their actual number. However, no marker of mitochondrial density or mitochondrial biogenesis has been measured in this study. Since exercise has been postulated as a stimulus inducing mitochondrial biogenesis in several organs/tissues, this statement sounds too speculative. The authors should better justify or remove at all this statement.   

Reviewer 2 Report

The authors present an interesting study on the effect of add-on 4-week moderate exercise training compared to usual training program in patients with previous stroke (>1 year) on aerobic capacity and platelet mitochondrial bioenergetics of importance for hemostasis.

The authors report an impressive improvement in systemic aerobe capacity as well as in the mitochondrial bioenergetics and a 100% compliance rate with no dropouts.

However, some major points need to be addressed:

Though some information is given on the patient group included, more is required to fully evaluate the importance of the results, as other factors may influence the platelet mitochondrial function. In particular, the authors should provide information on smoking, type of platelet inhibitor medication (ASA, clopidogrel, cilostazol, or dipyridamole?), type of statin. Further, more information on the subtype of the index stroke (cardioembolic, large artery, small vessel occlusion, other?) should be given. Please provide the rationale for including patients with intracerebral hemorrhage? A change in platelet function may have a detrimental effect in patients with ICH.

The authors evaluate three patients to have stroke symptom severity according to Brunnstrom stage II, which includes spasticity. Please provide information on how patients with spasticity were able to complete the required training program. Please provide information on the level of previous exercise in the patients prior to inclusion.

Please provide information on the nature and associated physical activity in the usual care training program.

The a priory power calculation should be provided, both on the primary endpoint and other relevant factors.

Please provide more information on blood sampling (time from last exercise session, handling of the blood samples) How did the authors ensure that the platelets were not activated prior to analysis? Please include possible pitfalls on platelet activation in pitfalls. Intense training may change platelet function per se.

Minor: Language revision would improve text in supplementary material.

Please revise line 342 page 11.